# The Influence of Burpee on Endurance and Short-Term Memory of Adolescents

**DOI:** 10.3390/ijerph191811778

**Published:** 2022-09-18

**Authors:** Georgiy Polevoy, Florin Cazan, Johnny Padulo, Luca Paolo Ardigò

**Affiliations:** 1Department of Physical Education, Moscow Aviation Institute, 125080 Moscow, Russia; 2Department of Physical Education, Moscow Polytechnic University, 125493 Moscow, Russia; 3Faculty of Physical Education and Sport, Ovidius University of Constanta, 900029 Constanța, Romania; 4Department of Biomedical Sciences for Health (SCIBIS), Università Degli Studi di Milano, 20133 Milan, Italy; 5Department of Teacher Education, NLA University College, 0166 Oslo, Norway; 6School of Exercise and Sport Science, Department of Neuroscience, Biomedicine and Movement Science, University of Verona, 37131 Verona, Italy

**Keywords:** exercise, distance-running, cognition, youth

## Abstract

Aim—The purpose of this study was to assess the effect of the Burpee exercise on the endurance and short-term memory of adolescents aged 15–16 years. Methods—The experiment was performed in a coeducational school in Kirov (Russia). The four-month study involved 52 adolescents of both genders. During the study period, 30 physical education lessons were held in each class. Adolescents from the control group were involved in a typical program (also aimed at improving endurance), and adolescents from the experimental group additionally performed the Burpee exercise. Endurance in adolescents was assessed by means of an “all-out” Running 2000 m test, and short-term memory was assessed by means of the Jacobs test (tests were performed before and after the programs). Results—An analysis of variance revealed an interaction effect (F = 28.733, η_p_^2^ = 0.578 and *p* < 0.001, and F = 104.353, η_p_^2^ = 0.676 and *p* < 0.001 for the Running 2000 m test and the Jacobs test, respectively). The control group improved by 1.9% (*p* > 0.05) in the Running 2000 m and by 5.5% (*p* > 0.05) in the Jacobs test. In the experimental group, both improved significantly by 8.6% (*p* < 0.05) in the Running 2000 m test and by 26.0% (*p* < 0.05) in the Jacobs test. Conclusion—The Burpee exercise could be included in physical education classes to improve endurance and short-term memory in 15–16-year-old.

## 1. Introduction

Almost everybody shares the aim of improving their physical and cognitive abilities. In particular, teachers and coaches are always in search of methods—including exercises—that are as effective as possible to achieve this aim. Among the most well-known specific exercises is the Burpee, which was invented by the America physiologist Royal Huddleston Burpee in New York in the late 1930s [1]. The Burpee exercise has become so popular that people from all over the world take part in online challenges held over the Internet with the goal of performing the maximum number of repetitions during a certain time interval. The Burpee is an exercise that places a particularly high burden on the metabolic machinery due to the large group of muscles involved in its practice [1]. The Burpee exercise can be performed anywhere and without any specific gym tool [2,3].

If performed with adequate technique, the Burpee exercise can support the safe development of the muscles of adolescents as it is easily performed with one’s own weight. Strength training in adolescence is an object of discussion because of its alleged risk to harm a person’s growth and development [4]. In fact, there is some controversy surrounding the appropriateness of performing resistance exercises in adolescents. There is a widely held inaccurate belief that strength training, when performed during puberty and/or adolescence, can hamper one’s growth by damaging growth plates. However, strength training is safe and does not negatively impact the growth and maturation of pre- and early-pubertal individuals [5,6]. Its benefits include an increase in strength, speed, and power; improved body composition; stronger bones; and a reduction in injury rates [7,8].

Despite its worldwide spread, there are no scientific studies showing the Burpee exercise as an effective element of physical activity for adolescents. It is likely that such an activity will be more effective within an individual approach for each student [9,10]. Furthermore, the scientific literature shows that physical activity has a positive effect on the cognitive performance of adolescents of different ages [11], in particular at 15–16 years of age, viz., when attending 10th grade or the first year of college [12,13,14].

Considering the sensitive periods of development of physical qualities such as strength, speed-strength abilities and endurance, the age of 15–16 years was determined to be best for the study (this corresponds to the 9th grade in regular school progression). At this age, there is a rather intensive development of the muscle mass of adolescents and notable development of most physical qualities, especially endurance [15,16]. If at primary school age, a differentiated approach and coordination training prevail, later, at secondary school age, an individual approach is more often used. The research hypothesis was that adding the Burpee exercise to traditional physical education lessons at school would lead to an improvement to both endurance and short-term memory. The aim of this study was to assess the effect of the Burpee exercise on the variables of endurance and short-term memory in adolescents aged 15–16 years. The implicit reasoning supporting our aim was that exercises such as the Burpee are different from many other single gym exercises. Rather, the Burpee exercise has, in our opinion, a double nature (read Section 2.3
*Execution Technique* below). Namely, the Burpee exercise appears to be particularly demanding of metabolic energy due to its fast-paced jumping and weight-bearing stages. Furthermore, its correct execution requires continuous and conscious motor control over its different stages, which are very different from each other and involve all the main body segments (thus, maybe, in some way, mirroring short-term memory capability [17]). It is noteworthy to recall that Burpee training was already deemed to be associated with more positive psychological responses than other commonly practiced high-intensity exercises [18]. Therefore, we expected that matching training time between an experimental group and a control group and adding the Burpee exercise to a usual physical education program would further improve both endurance and short-term memory.

## 2. Materials and Methods

### 2.1. Participants

Fifty-two adolescents aged 15–16 years (height 167.4 ± 8.9 cm, mass 56.2 ± 8.5 kg), who studied in a regular Russian school in the city of Kirov took part in this study. They were boys and girls who studied at grade 9 (in two different classes: 9a and 9b). During the study period, physical education classes were set so that other physical activities of the students did not affect the test results. That is, the students who took part in the study did not exercise additionally. The students were divided into a control group (CG) and an experimental group (EG). Regardless of gender and level of physical fitness, only those adolescents who were admitted to physical education lessons by a doctor without any of the following restrictions for health reasons took part in the study. Students’ non-inclusion *criteria* were: joint problems (especially at knee and hip), chronic heart disease or high blood pressure, and being excessively overweight (more than 30% of the normal weight). In total, out of 32 students studying in 9a, 25 completely healthy—as assessed by the school’s physician—students (13 boys and 12 girls) were admitted to the study as the CG. In the same way, out of 33 students studying in 9b, 27 students (13 boys and 14 girls) took part in the study as the EG. All procedures met the ethical standards of the 1964 Declaration of Helsinki and were approved by the local university ethics committee. Informed consent was obtained from all the parents of the adolescents included in the study. Students were free to withdraw from the experiment at any time.

### 2.2. Procedure

Research was performed in a co-educational school in Kirov (Russia) from 1 September to 30 December 2021. School physical education lessons took place twice a week, lasting 45 min each. All lessons were monitored. A total of 30 physical education lessons were performed, always at the same time and on the same weekdays. A usual physical education program for students at grades 1–11 was administered to the CG [19]. The purpose of the physical education curriculum was to form the basics of a healthy lifestyle among school students and the development of creative independence through the development of motor activity. The realization of the purpose of the curriculum correlated with the solution of the following educational tasks:Improving knowledge on the importance of physical culture for strengthening human health and on its positive impact on human development (physical, intellectual, emotional and social).Strengthening the health of schoolchildren through the development of physical qualities and increasing the functionality of the life-supporting systems of the body.Improving vital skills and abilities through learning outdoor games, physical exercises and technical actions from basic sports.

The experimental group followed the same program but additionally performed the Burpee exercise in each lesson (in Table 1, differences from the control group, which was engaged in the ordinary program for the whole duration of each lesson, can be observed; i.e., lessons probably resulted in a few differences among the groups in terms of training volume).

### 2.3. Execution Technique

If we split the whole Burpee exercise into separate stages, then it consists of four main simple exercises: squat, plank, push-up and jumping up. In more detail, the technique of the traditional Burpee exercise is as follows [20]:Place the hands approximately shoulder-width apart.Jump back to a push-up position.Lower chest and thighs to the ground.Push up and jump the feet back toward the hands.Jump to full hip and knee extension.Extend the arms overhead during the jump.

Particular attention was paid to performing the correct technique rather than aiming at a high number of repetitions. Nevertheless, a sufficiently high intensity was required to obtain a significant effect. In case of dizziness, nausea or chest pain, the exercise had to be stopped.

### 2.4. Exercise Stress

The exercise was performed for a given time, and an operator counted the repetitions of each student (from 15 to 25 repetitions per minute).

The Burpee exercise was performed at the end of the main part of the lesson. Overtime, exercise administration followed the principle of gradualism [21]. The administration features of the Burpee exercise over time are shown in Table 1.

It should be noted that the Burpee exercise administration features were chosen considering the well-being of the majority of adolescents involved. For example, during the first month, a 60 s period of the Burpee exercise was ineffective, but a longer exercise time was not tolerated by most of the students. Gradually, the Burpee exercise time increased, and the rest time decreased. However, at the end of the experimental period, most of the students could not physically endure more than three series, as described above. At the same time, it was important that the Burpee exercise did not interfere with the implementation of the regular main physical education curriculum at the school.

### 2.5. Test Procedure

In the last lesson before the start of the study period and in the first lesson after the end of the study period, all adolescents took two control tests. General endurance was assessed by means of the Running 2000 m test [19]. The test consisted of running 2 km on the track of the athletics arena. Before the start of the test, the students were in a standing position. At the command “Start!”, the students took their place behind the starting line. At the command “March!”, they started running. The results were recorded using a chronometer in minutes and seconds with an accuracy of 1 s.

Short-term memory was determined by means of the Jacobs test [22]. This well-known test has been repeatedly positively assessed over time for its validity and reliability [23]. There were four squares on a paper sheet. Each square had numbers with figures in random order and numbers of digits in ascending order over each column (Table 2).

In short, the test was as follows: the teacher slowly called aloud the figures of each line (one square after the other starting from the first). After each line, the adolescents wrote the numbers in the order in which the teacher told them. Short-term memory scores were calculated using the formula A + C/4, where A was the digit number of the longest line that was written correctly, and C was the number of correctly written lines shorter than the longest correctly written. The reason for the teacher to read all the numbers in all the squares was to provide the adolescents with the greatest number of opportunities to remember the numbers. For example, if an adolescent wrote all the first lines (A = 4), they received a 4-point score (C = 0, A + C/4 = 4).

### 2.6. Statistical Processing of Results

The results were recorded as means and standard deviations. Statistics was performed with SPSS 20.0 (IBM, Armonk, NY, USA). Normality of results was assessed and confirmed by means of the Shapiro–Wilk test (threshold with *p* < 0.05). The null hypothesis for the values was assessed by means of the multivariate Pillai test. *T*-test was used to compare the two groups for the Running 2000 m test and Jacobs test at baseline conditions. The between-group differences in Running 2000 m test and Jacobs test over-time were analyzed using a two-way analysis of variance (ANOVA) with time as a repeated-measure factor (two levels, pre- and post-training) and group as a between factor (two levels, CG and EG) with Bonferroni post hoc test, if the sphericity was not violated. The ANOVA effect size was evaluated with partial eta squared (η_p_^2^) and classified as follows: small, <0.06; medium, 0.06–0.14; and large, >0.14 [24]. The magnitude of differences between variables was interpreted using standardized effect size (Cohen’s *d*): <0.1, no effect; 0.20 < 0.40, small effect; 0.50 < 0.70, intermediate effect; 0.80 < 1.0, large effect. Moreover, upper and lower 95% confidence intervals of the difference (95% CI) were calculated. The statistical significance level was set at *p* < 0.05.

## 3. Results

The Running 2000 m test and Jacobs test initial values were similar (*p* > 0.05). The drop-out rate was nil. First, the Shapiro—Wilk test showed a normal distribution. For the Running 2000 m test, *p* = 0.918 with W = 0.940 and *p* = 0.918 with W = 0.942 for the CG and the EG, respectively. Similarly for the Jacobs test, *p* = 0.918 with W = 0.936 and *p* = 0.918 with W = 0.959 for the CG and the EG, respectively. The sphericity was not violated. At baseline conditions, no significant differences were found between groups for the Running 2000 m test (*p* = 0.471) and the Jacobs test (*p* = 0.063). All data are reported in Table 3.

Then, the multivariate Pillai test was used to preliminarily test the null hypothesis for the values. The test results are V = 0.635 with *p* < 0.001 and V = 0.676 with *p* < 0.001 for the Running 2000 m test and the Jacobs test, respectively. An analysis of variance was operated to assess interaction effects. The analysis of variance revealed interaction effects of F = 28.733, η_p_^2^ = 0.578 and *p* < 0.001, and F = 104.353, η_p_^2^ = 0.676 and *p* < 0.001 for the Running 2000 m test (*d* = large effect) and Jacobs test (*d* = large effect), respectively.

As shown in Table 3, the performances in both tests improved significantly only in the EG. In adolescents of the CG, the scores in the Running 2000 m test improved by 1.9% (*p* > 0.05, not significantly) and the scores in the Jacobs test increased by 5.5% (*p* > 0.05, not significantly) compared to before the start of the study period. Differently, for adolescents in the EG, the scores in the Running 2000 m test improved by 8.6% (*p* < 0.05), and the scores in the Jacobs test increased by 26.0% (*p* < 0.05) in respect to before the intervention.

## 4. Discussion

First, this study unexpectedly highlighted the lack of effectiveness of the administered physical education program to increase running endurance, as demonstrated by the results in the control group (CG) shown in Table 3. It was not expected that such a program would specifically increase short-term memory, and this was revealed effectively to be the case. However, as hypothesized, adding the Burpee exercise to the chosen program (maintaining the same lesson’s duration) improved both endurance and memory. We do not know whether the Burpee exercise caused these positive effects per se or if it was the interaction with the program. This matter could be further studied. Certainly, adding the Burpee to an exercise program constitutes at least a preliminary practical approach to improving the effectiveness of physical education programs such as the one that was adopted.

Regarding the comparison between our results and the existing literature, Table 4 shows data intervals for adolescents aged 15–16 years in terms of endurance variables in the Running 2000 m test [19] and short-term memory in the Jacobs test [22].

Table 4 shows that from the beginning to the end of the study, the variables of adolescents in the CG in the Running 2000 m test remained within the medium result range, whereas for the adolescents from the EG, the variables improved from poor to good results [19]. Regarding the Jacobs test, the variables of short-term memory in adolescents from the CG changed from poor to medium results, whereas in the EG, the variables improved from poor to excellent results [22].

Teachers and coaches always look for the optimal times to develop the abilities of their students and athletes [11]. The Burpee exercise appears to be almost unique given its lack of need for any large space or specific gym tool while involving many muscles [2,3]. Overall, the exercise improves endurance in adolescents aged 15–16 years [19]. Different studies have confirmed this age as particularly sensitive for endurance improvement [4,15].

Physical activity should be individual, especially at senior school age [9,16]. In our research, a certain amount of activity time for the Burpee exercise in adolescents aged 15–16 years was administered during physical education lessons at school, and its effect on endurance and short-term memory was assessed. It is likely that administering physical activity for a time shorter than the amount chosen within our research would not be that effective. In turn, if more time was allocated to the Burpee exercise, then the adolescents would risk becoming too physically and psychologically tired and lose interest in their exercise, and/or there may not be enough time to follow the main physical education teachings at the school [19]. It is noteworthy to remember that the effect of exercise on brain function in adolescents is dose-dependent [25], and, therefore, future studies should focus on the dose of an exercise, e.g., the Burpee exercise, in daily life for improving cognition among adolescents.

Some research has highlighted the positive impact of physical activity on mental and cognitive abilities [12,13,14,26]. Our findings regarding adolescents from the EG confirm these data, since their memory variables improved significantly after the study period, and, namely, there was a significant improvement in the Jacobs test. This well-known test has been repeatedly positively assessed over time for its validity and reliability [26]. In the same way, adolescents from the EG significantly improved their endurance variables, which once again confirms the efficacy of the introduction of the Burpee exercise into the educational path of adolescents aged 15–16 years [27]. Thus, the proposed hypothesis was verified.

The design of the present research could be further improved by exactly matching the EG training volume with the CG. Future studies could further investigate the Burpee exercise (and/or other similar exercises), in particular its positive link to the improvement of short-term memory shown in this study. 

## 5. Conclusions

When the physical exercise Burpee was performed during each lesson of physical education at school, the endurance variables of adolescents aged 15–16 years improved significantly. Likewise, variables of cognitive capability, such as memory, also improved significantly. The present research provides an example of how physical activity is useful for students in the 9th grade of a regular school.

## Figures and Tables

**Table 1 ijerph-19-11778-t001:** Over-time Burpee exercise administration features for adolescents aged 15–16 years.

Number of Series	Activity	September	October	November	December
1st series	Burpee	60 s	90 s	120 s	120 s
	Time of rest	60 s	60 s	90 s	60 s
2nd series	Burpee	60 s	90 s	120 s	120 s
	Time of rest	90 s	90 s	120 s	90 s
3rd series	Burpee	60 s	90 s	90 s	120 s
	Time of rest	120 s	120 se	120 s	120 s
Total time		7 m 30 s	9 m	11 m	10 m 30 s

**Table 2 ijerph-19-11778-t002:** Material for determining short-term memory.

First Square	Second Square
7428	7364
96538	95837
694103	104735
3620395	0987295
20315631	03827547
738602126	930472659
4839039516	9408726351
**The third square**	**Fourth square**
5829	9813
73620	93846
831965	019462
0946283	9273657
72840586	62539506
637495024	726304957
1048375629	7076241546

**Table 3 ijerph-19-11778-t003:** The tests results for all groups.

Test	Control Group (n = 25)					Experimental Group (n = 27)				
	Before	After	%	*d*	*p*	Before	After	%	*d*	*p*
Running 2000 m (min.s)	9.95 ± 1.16	9.76 ± 1.04	−1.9 ± 1.8	Large	*p* > 0.05	10.19 ± 1.21	9.27 ± 0.67	−8.6 ± 5.0	Large	***p* < 0.05**
Jacobs (pt)	6.2 ± 0.6	6.5 ± 0.4	5.5 ± 3.1	Large	*p* > 0.05	5.9 ± 0.4	7.4 ± 0.5	26.0 ± 10.3	Large	***p* < 0.05**

**Table 4 ijerph-19-11778-t004:** Data intervals of Running 2000 [19] and Jacobs [22] for adolescents aged 15–16 years.

Evaluation	Running 2000 m (min.s)	Jacobs (pt)
Excellent result	9.00–9.20	7.4–7.7
Good result	9.21–9.40	6.8–7.3
Medium result	9.41–10.00	6.3–6.7
Poor result	10.01–10.20	5.6–6.2
Very bad result	Below 10.20	Below 5.6

## Data Availability

The data presented in this study are available on request to the corresponding author.

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
