# Peer review of "The Influence of Burpee on Endurance and Short-Term Memory of Adolescents"

_ijerph, 2022, doi:10.3390/ijerph191811778_

Round 1
Reviewer 1 Report (Previous Reviewer 1)
First of all, I would like to thank the authors for the effort they have made to improve the description of the study. However, I would like the authors to take into account the following considerations:
- The introduction lacks the contextualization of why the short-term memory variable will be studied. What relationship is established?
- There isn't the title of "Methods"
- In the line 243 delete the phrase referring to the normality test since it has been explained in the statistical analysis.
- If the sessions had the same duration in both groups, fewer activities/exercises were performed in the EG than in the CG, as described in the limitations. Thus, it would be necessary to describe more clearly what the differences were in the sessions.
- Renumber tables, there are two "table 3"
- The statistical test to show that the two groups were equal before the start of the study is missing.
Author Response
Response to Reviewer 1 Comments
Does the introduction provide sufficient background and include all relevant references?
(x) Can be improved
Please read below comments to specific points.
Are the methods adequately described?
(x) Can be improved
Please read below comments to specific points.
Are the results clearly presented?
(x) Can be improved
Please read below comments to specific points.
Are the conclusions supported by the results?
(x) Can be improved
Please read below comments to specific points.
Point 1: - The introduction lacks the contextualization of why the short-term memory variable will be studied. What relationship is established?
Response 1: We thank expert reviewer for her/his suggestion. Short-term memory assessment contextualization was stated as well by adding one further reference as follows:
“Furthermore, its correct execution requires continuous and conscious motor control over its different stages, which are very different each other and involving all the main body segments (and thus, maybe, in some way mirroring short-term memory capability [17]).”
Point 2: - There isn't the title of "Methods".
Response 2: Typo was amended.
Point 3: - In the line 243 delete the phrase referring to the normality test since it has been explained in the statistical analysis.
Response 3: Sentence was changed as follows:
“First of all, Shapiro-Wilk test showed a normal distribution.”
Point 4: - If the sessions had the same duration in both groups, fewer activities/exercises were performed in the EG than in the CG, as described in the limitations. Thus, it would be necessary to describe more clearly what the differences were in the sessions.
Response 4: As acknowledged as a study limitation, training volume was not assessed. Both groups performed physical activity for the same time, but experimental group performed the Burpee exercise during a limited lesson’s time while the control kept performing the ordinary activity. However, sentence was changed as follows:
“Experimental group was administered the same program, but additionally per-formed the Burpee exercise at each lesson (differently from the control group, which was engaged in the ordinary program over the whole duration of each lesson, i.e., lessons resulted probably a little different among groups in terms of training volume).”
Point 5: - Renumber tables, there are two "Table 3".
Response 5: Typo was amended.
Point 6: - The statistical test to show that the two groups were equal before the start of the study is missing.
Response 6: We thank expert reviewer for his suggestion. Information was provided.
We hope that the manuscript has now reached the standard necessary for formal acceptance endorsement in International Journal of Environmental Research and Public Health.
We look forward to hearing from you.
Best regards
Reviewer 2 Report (New Reviewer)
Dear authors,
the paper has a good scientific soundness as the theme of an effective physical education lesson is of paramount interest. However, in my opinion, there are several issues that must be addressed in order to further proceed with publication. I would also recommend a thorough English revision with a mother language approach and avoiding colloquial style. Also, the formatting style must adhere to IJERPH indications for authors.
In the pdf file, you will find all my notes and the points that I have raised.
Kind regards

Author Response
Response to Reviewer 2 Comments
Does the introduction provide sufficient background and include all relevant references?
(x) Can be improved
Please read below comments to specific points.
Are all the cited references relevant to the research?
(x) Can be improved
Please read below comments to specific points.
Is the research design appropriate?
(x) Can be improved
Please read below comments to specific points.
Are the methods adequately described?
(x) Must be improved
Please read below comments to specific points.
Are the results clearly presented?
(x) Must be improved
Please read below comments to specific points.
Are the conclusions supported by the results?
(x) Can be improved
Please read below comments to specific points.
Point 1: I would also recommend a thorough English revision with a mother language approach and avoiding colloquial style.
Response 1: We thank the expert reviewer for her/his suggestion. Manuscript was further professionally proofread by an academic proofreading service.
Point 2: Also, the formatting style must adhere to IJERPH indications for authors.
Response 2: Manuscript was prepared starting from the IJERPH template. Some relevant typos were operated (inadvertently) by Editorial Manager.
Point 3: (line 14) the purpose
Response 3: Suggestion was operated.
Point 4: (l14) Strikethrough Text
Response 4: Suggestion was operated.
Point 5: (l25) Highlighted Text
Response 5: Suggestion was operated.
Point 6: (l29) Highlighted Text
Response 6: Suggestion was operated.
Point 7: (l32) please rephrase this, it is too colloquial.
Response 7: Suggestion was operated.
Point 8: (l33) are always in search for methods
Response 8: Suggestion was operated.
Point 9: (l43) one’s own weight
Response 9: Suggestion was operated.
Point 10: (l52) there are no scientific
Response 10: Suggestion was operated.
Point 11: (l53) element of physical
Response 11: Suggestion was operated.
Point 12: (l58÷81) Why is all this long paragraph in bold?
I would also suggest to conclude the paragraph with the aim of the study after having presented the research hypothesis and the related previous studies.
Response 12: Typo was amended. Suggestion was operated.
Point 13: (l59) aged
Response 13: Suggestion was operated.
Point 14: (l63) is
Response 14: Suggestion was operated.
Point 15: (l66÷9) please rephrase this: “the research hypothesis was that adding Burpee exercise to traditional physical education lesson at school would have led to improvement to both endurance and short-term memory.”
Response 15: Suggestion was operated.
Point 16: (l81) place this properly
Response 16: Typo was amended.
Point 17: (l83) 52 adolescents aged
Response 17: Suggestion was operated.
Point 18: (l84-5) Strikethrough Text
Response 18: Suggestion was operated.
Point 19: (l87) Strikethrough Text
Response 19: Suggestion was operated.
Point 20: (l87) so that
Response 20: Suggestion was operated.
Point 21: (l87-8) Indeed, the students
Response 21: Suggestion was operated.
Point 22: (l106-7) a usual physical education program for students at grades 1-11 was administered to CG
Response 22: Suggestion was operated.
Point 23: (l119) followed the
Response 23: Suggestion was operated.
Point 24a: (l119-20) how many reprs for how many sets or for how long? here authors should specify or refer to Table 1.
Point 24b: (l212) then authors should clarify this point also in the methods section. which part was then cut for the EG?
Point 24c: (l314) as above suggested
Point 24d: (l315) idem
Response 24: We referred to Table 1. Sentence was clarified.
Point 25: (l121) check the style. here it is the italic and it should be moved to the next line.
Response 25: Typo was amended.
Point 26: (l144) too colloquial. therefore is more appropriate
Response 26: Suggestion was operated.
Point 27: (l206) respect to before the intervention
Response 27: Suggestion was operated.
Point 28: (l208-9) in which group? it is unclear
Response 28: Sentence was clarified.
Point 29: (l212) please avoid the form “we”. Impersonal form is always preferrable
Response 29: Suggestion was operated.
Point 30: (l216) as above suggested
Response 30: Suggestion was operated.
Point 31: (l230) aged
Response 31: Suggestion was operated.
Point 32: (l233) please be sure to correct all of these
Response 32: Suggestion was operated.
Point 32: (l235) as above suggested
Response 32: Suggestion was operated.
Point 33: (l235) unclear, please be more precise.
Response 33: Sentence was clarified.
Point 34: (l246-7) move this sentence to the methods section
Response 34: Suggestion was operated.
Point 35: (l250) Strikethrough Text
Response 35: Suggestion was operated.
Point 36: (l253) Strikethrough Text
Response 36: Suggestion was operated.
Point 37: (l253÷5) unclear. this was one the aims of your study, please be more clear.
Response 37: Sentence was clarified.
Point 38: (l260) the present research
Response 38: We thank expert reviewer for his suggestion. Suggestion was operated.
We hope that the manuscript has now reached the standard necessary for formal acceptance endorsement in International Journal of Environmental Research and Public Health.
We look forward to hearing from you.
Best regards
Round 2
Reviewer 2 Report (New Reviewer)
Dear authors,
I would like to thank you for your responses. However, I feel that sources such as Wikipedia and YouTube are not entirely adequate for a reference list of a high-level journal.
Could you please provide more scientific ones?
Author Response
Response to Reviewer 2 Comments
Are all the cited references relevant to the research?
(x) Can be improved
Please read below comment to specific point.
Point 1: … However, I feel that sources such as Wikipedia and YouTube are not entirely adequate for a reference list of a high-level journal.
Could you please provide more scientific ones?
Response 1: We thank expert reviewer for her/his suggestion. I provided more scientific references.
We hope that the manuscript has now reached the standard necessary for formal acceptance endorsement in International Journal of Environmental Research and Public Health.
We look forward to hearing from you.
Best regards
This manuscript is a resubmission of an earlier submission. The following is a list of the peer review reports and author responses from that submission.
Round 1
Reviewer 1 Report
Dear authors,
This article aims to evaluate the effects of performing burpees in addition to Physical Education sessions. However, there are some aspects that should be reconsidered:
- Introduction: the theoretical framework should be expanded.
- Methodology: describe the objectives of the Physical Education sessions.
- Results: They should be rewritten, in addition to organizing the content and tables.
- Discussion: it would be necessary to expand it. For example, it could be analyzed if the baseline and final correspond to adolescents of their age, for both tests. In addition, add other similar studies to corroborate or give another opinion regarding the changes obtained. And justify why it is believed that improvements in short-term memory have been obtained.
Thank you very much.
Reviewer 2 Report
The present paper investigated the effects of burpee added to an undefined usual physical education program. It concluded burpee is efficient to increase endurance and short-term memory. The quality of the paper is relatively low with numerous important issues (such as the control condition). The rationale is relatively unclear and authors consider the exercise like a training modality whereas it is just a particular global exercise conducted in such a way it will develop muscle endurance. Moreover, the discussion is poor without any mechanistic arguments.
Citation (1st page, bottom, left hand side) should be completed
Abstract, first line. Delete 1 aim
Abstract, line 16. Include the gender.
Abstract line 17. What is an usual program (not in detail but main aim). If the usual program is not aimed to improve endurance, the study is directly biased.
Abstract, line 19. All-out running test?
Abstract, line 19. Should indicate that test were performed before and after the training program.
Abstract, line 20. What is a group interaction effect?
Keywords should not replicate title words. Why crossfit and individual approach for keywords. Authors should modified all keywords.
Introduction. This part is unclear. Authors need to had references and authors need to clearly develop the rationale: why this exercise could be better than others? Is it related to the exercise? Is it related to the fact that individuals train muscle endurance? Because the introduction does not clearly explain the advantage of the type of training (power vs. strength vs. hypertrophy vs. endurance) or the type of exercise (analytic vs. global vs. unilateral vs. bilateral), the rationale could not be understood. To summarize, the introduction is in a single direction (in favor of burpee) while this practice (I use the term practice to indicate it is multicomponent: exercise and endurance) could be performed with other type of exercises that involve the entire body using a similar muscle endurance type.
Introduction. Why words in italic
Introduction, line 33. A reference for this affirmation please.
Introduction, L37. What does « accelerated the work of the heartbeat » mean? Nothing for me.
Introduction, line 39. Please add a reference. Other general exercises also involve numerous muscle groups.
Introduction, Line 41. Without a trainer is a shame. This exercise is often performed with inadequate technique and efficiency. The question of the safety is open (line 45).
Introduction, Line 46-48. Barbells and dumbells do not mean unsafe movements (ie loads could be lower than body weight). Please precise.
Introduction Line, 50-57. This sentence has no link with the preceding sentences. It should move at the end of introduction to justify the choice of this age group.
Introduction Line 58. Part of this sentence is redundant.
Introduction, Line 61-64. This aspect should be developed for clarity. The reader has difficulties to find the link between exercise, training, exercise type and cognition. Accordingly, it is not clear why authors hypothesize physical activity program including burpee could be more efficient for short-term memory than other exercises (L67-69).
Methods.
Methods, line 82. Was the study approved by an ethics committee?
Methods, line 91-92. We don’t know what a usual physical education program is. It should be described in detail to indicate the aim (sport, physical activity, which physical quality is developed…) and exercises used (running, strengthening…).
Methods, Line 92-93. Rephrase please.
Methods, Line 95-107. Add a picture and if possible references.
Methods, Line 108-115. Why this part? should we interpret this part as exclusion or non-inclusion criteria? What’s happening if volunteers had to stop the exercise? Were they excluded from the study? Same question for the usual physical program. Also, what is the minimum number of training sessions to be included is statistical analyses? Were all training sessions monitored? What are volunteers performing outside of the school?
Methods, Line 116-124. What does « administered on an individual basis mean »? Why at different part of the training (warm-up or end of session)? For what aim? The administration of this exercise is unclear.
Methods, Line 134. When exactly were conducted the tests? It is written before and after the start. What is before? and what is after the start? Don’t you have post-tests?
Methods, Line 140-149. The test is unclear. I don’t understand the square and figures…
Methods. It is unclear why these tests were performed. For instance, running was not trained, memory was not trained. Burpee was trained but not evaluated.
Methods, Line 150. Did authors controlled the sphericity?
Results.
Results, Line 170. Pillai test? It is not indicated in statistics.
Results, L174-175. I disagree with this sentence. According to table 3, tests were not improved in CG.
Results, Table 3. Percentage values have been calculated with the mean values and have not been calculated with individual data. These values are false. Please calculated the true percentage change and give the standard deviation.
Results. The so-calculated percentage changes could be tested using a student t-test to determine whether values were effectively significantly different. As conducted in this paper, we can not conclude the gains were different.
Results. Were initial values similar between groups?
Discussion.
The discussion does not clearly discussed the study results and does not clearly and exhaustively discuss the processes (physiological and cognitive processes). As indicated earlier, we don’t know what is a usual physical education program and this program is not discuss in details. As indicated earlier, the study is conducted with an a priori and exclude any possible interaction between other components of training. Is endurance type effort effective? or is this specific exercise effective? The other main issue is that groups were not matched. For instance, a CG performing another endurance type exercise could have the same gains… As indicated earlier, the link between exercise and memory is unclear. Also, we don’t understand why the usual physical education program don’t produce any increase (in running endurance or memory).